# Abnormal Calcium Handling in Atrial Fibrillation Is Linked to Changes in Cyclic AMP Dependent Signaling

**DOI:** 10.3390/cells10113042

**Published:** 2021-11-05

**Authors:** Franziska Reinhardt, Kira Beneke, Nefeli Grammatica Pavlidou, Lenard Conradi, Hermann Reichenspurner, Leif Hove-Madsen, Cristina E. Molina

**Affiliations:** 1Department of Cardiovascular Surgery, University Heart & Vascular Center Hamburg UKE, 20251 Hamburg, Germany; fr.reinhardt@uke.de (F.R.); l.conradi@uke.de (L.C.); reichenspurner@uke.de (H.R.); 2German Center for Cardiovascular Research (DZHK), Partner Site Hamburg/Kiel/Lübeck, 20251 Hamburg, Germany; k.beneke@uke.de (K.B.); nefeligramm@gmail.com (N.G.P.); 3Institute of Experimental Cardiovascular Research, University Medical Center Hamburg-Eppendorf (UKE), 20251 Hamburg, Germany; 4Biomedical Research Institute Barcelona, IIBB-CSIC and IIB Sant Pau, Hospital de la Santa Creu i Sant Pau, 08025 Barcelona, Spain; leif.hove@iibb.csic.es

**Keywords:** atrial fibrillation (AF), cAMP-dependent regulation, protein kinase A (PKA), L-type calcium current (I_Ca,L_), transient inward current (I_TI_), patch-clamp

## Abstract

Both, the decreased L-type Ca^2+^ current (I_Ca,L_) density and increased spontaneous Ca^2+^ release from the sarcoplasmic reticulum (SR), have been associated with atrial fibrillation (AF). In this study, we tested the hypothesis that remodeling of 3′,5′-cyclic adenosine monophosphate (cAMP)-dependent protein kinase A (PKA) signaling is linked to these compartment-specific changes (up- or down-regulation) in Ca^2+^-handling. Perforated patch-clamp experiments were performed in atrial myocytes from 53 patients with AF and 104 patients in sinus rhythm (Ctl). A significantly higher frequency of transient inward currents (I_TI_) activated by spontaneous Ca^2+^ release was confirmed in myocytes from AF patients. Next, inhibition of PKA by H-89 promoted a stronger effect on the I_TI_ frequency in these myocytes compared to myocytes from Ctl patients (7.6-fold vs. 2.5-fold reduction), while the β-agonist isoproterenol (ISO) caused a greater increase in Ctl patients (5.5-fold vs. 2.1-fold). I_Ca,L_ density was larger in myocytes from Ctl patients at baseline (*p* < 0.05). However, the effect of ISO on I_Ca,L_ density was only slightly stronger in AF than in Ctl myocytes (3.6-fold vs. 2.7-fold). Interestingly, a significant reduction of I_Ca,L_ and Ca^2+^ sparks was observed upon Ca^2+^/Calmodulin-dependent protein kinase II inhibition by KN-93, but this inhibition had no effect on I_TI_. Fluorescence resonance energy transfer (FRET) experiments showed that although AF promoted cytosolic desensitization to β-adrenergic stimulation, ISO increased cAMP to similar levels in both groups of patients in the L-type Ca^2+^ channel and ryanodine receptor compartments. Basal cAMP signaling also showed compartment-specific regulation by phosphodiesterases in atrial myocytes from 44 Ctl and 43 AF patients. Our results suggest that AF is associated with opposite changes in compartmentalized PKA/cAMP-dependent regulation of I_Ca,L_ (down-regulation) and I_TI_ (up-regulation).

## 1. Introduction

Atrial fibrillation (AF) has been commonly associated with profound remodeling of calcium (Ca^2+^)-handling and regulatory proteins [1,2,3,4]. Thus, AF-myocytes displayed a reduced L-type Ca^2+^ current (I_Ca,L_) density [2,5,6], as well as an increased spontaneous sarcoplasmic reticulum (SR) Ca^2+^ release [1] through the ryanodine receptor (RyR2) called Ca^2+^ sparks and Ca^2+^ waves. The increased Ca^2+^ waves then resulted in a rise in concurrent transient inward current (I_TI_), due to Ca^2+^ extrusion by the Na^+^-Ca^2+^ exchanger (NCX) [7], and the corresponding additional depolarization of the membrane.

A common mechanism proposed to explain Ca^2+^-handling remodeling is a change in the phosphorylation state of the related proteins [3,8,9,10]. Two distinct serine residues of RyR2 have displayed increased phosphorylation in AF. The first one, ser2809 (ser2808 in rodents) [3,5,6], has been shown to be phosphorylated by protein kinase A (PKA). 3′,5′-Cyclic adenosine monophosphate (cAMP)-dependent activation of PKA is indeed a major pathway for modulation of protein phosphorylation. cAMP levels are in turn regulated by adenylyl cyclases (AC), which generate cAMP upon G protein-coupled receptor stimulation by catecholamines, and by phosphodiesterases (PDEs), which degrade cAMP and regulate its propagation, mediating compartmentalized cAMP signals [4]. Additionally, the reduced I_Ca,L_ density has been shown to be in part linked to a phosphatase-dependent reduction in the phosphorylation state of the L-type Ca^2+^ channel [11]. The second one, ser2815 (ser2814 in rodents), has been shown to be phosphorylated by Ca^2+^/Calmodulin-dependent protein kinase II (CaMKII) [6,10]. CaMKII activity is actively regulated by Ca^2+^, calmodulin and phosphatases. Furthermore, intracellular increase in cAMP levels can also activate a guanine nucleotide exchange factor, the exchange protein activated by cAMP (Epac) [12], and phosphorylate RyR2 via CaMKII in a PKA-independent manner [13].

Therefore, remodeling of the phosphorylation state of Ca^2+^ regulatory proteins is emerging as an important mechanism modulating abnormal Ca^2+^ handling in AF. However, these findings suggest opposite modulatory phosphorylation states between the RyR2 [3,5] and the L-type Ca^2+^ channel [9], making it important to determine if this is due to alterations or compartmentalization of the regulating phosphorylation pathways.

This study, therefore, tested the hypothesis that the remodeling of L-type Ca^2+^ current and SR Ca^2+^ channel activity in AF is due to changes in cAMP-dependent PKA signaling, the key pathway for β-adrenergic receptor-mediated regulation of the phosphorylation state of these channels. Hence, the effect of the pharmacological manipulation of cAMP-dependent signaling on I_Ca,L_ density, frequency of SR Ca^2+^ release events large enough to produce I_TI_ and cAMP levels was investigated in atrial myocytes from patients in sinus rhythm (Ctl) and with AF. Specifically, PKA and CaMKII were inhibited to eliminate PKA and CaMKII-dependent Ca^2+^ channel phosphorylation, and β-adrenergic signaling was stimulated to maximize Ca^2+^ channel phosphorylation.

Our data show that in AF, PKA-inhibition with H-89 had little effect on the I_Ca,L_ amplitude, but strongly reduced spontaneous Ca^2+^ release large enough to produce spontaneous transient inward currents. These results suggest that, opposite to sinus rhythm, PKA-dependent phosphorylation is minimal for the L-type Ca^2+^ channel but eminent for the RyR2 at baseline in AF patients. However, β-adrenergic stimulation with isoprenaline increased I_Ca,L_ amplitude, I_TI_ and cAMP levels in all compartments in both groups of patients. Therefore, changes in Ca^2+^ handling seen in patients with AF are likely due to modulations in cAMP-dependent regulation.

## 2. Materials and Methods

### 2.1. Human Atrial Samples

A total of 244 atrial human tissue samples were collected from patients undergoing cardiac surgery at the University Heart and Vascular Center Hamburg-Eppendorf (UKE). Atrial tissues were taken from patients in sinus rhythm (Ctl) or with atrial fibrillation (AF), subjected to atrial cannulation for extracorporeal circulation. All samples were taken with informed consent of the donors. The study was conducted in accordance with the Declaration of Helsinki principles and approved by the Ethical Committees of the Ärztekammer Hamburg (Ethical approval Number: WF-088/18, 26 February 2019).

Patients treated with Ca^2+^ antagonists were excluded from the study. Overall patient characteristics, including diagnostics and treatment were summarized in Table 1.

### 2.2. Isolation and Culture of Human Atrial Myocytes

A total of 428 human atrial myocytes from 148 patients in sinus rhythm and 96 patients with AF were isolated as previously described [14,15] and used for experiments in this study. Right atrial appendage tissue samples, normally discarded during surgery, were cut into small pieces of tissue, and incubated at 35 °C in a Ca^2+^ free solution containing 0.5 mg/mL collagenase (Worthington type 2, 323 U/mg; Lakewood, NJ, USA), 0.5 mg/mL proteinase (Sigma type XXIV, 11 U/mg solid; St. Louis, MO, USA) and 2% bovine serum albumin (BSA; Sigma, St. Louis, MO, USA). After 30 min, the tissue was removed from the enzyme solution, and cells were disaggregated with a Pasteur pipette in Ca^2+^-free solution containing 5% BSA and 2 µM blebbistatin (Sigma, St. Louis, MO, USA). The remaining tissue was digested for 3 × 15 min in a fresh Ca^2+^ free solution containing 0.4 mg/mL collagenase and 2% BSA. Freshly isolated cells were suspended in Ca^2+^-free solution containing 5% BSA for patch-clamp experiments (duration ≈ 8 h) or in minimal essential medium (MEM: M 4780; Sigma, St. Louis, MO, USA) containing 1.2 mmol/L Ca^2+^, 2.5% fetal bovine serum (FBS, Invitrogen, Cergy-Pontoise, France), 1% penicillin-streptomycin, 2% HEPES and 2 µM blebbistatin (pH 7.6) and plated on 35 mm, laminin-coated culture dishes (10 μg/mL laminin, 2 h; Sigma, St. Louis, MO, USA) for fluorescence resonance energy transfer (FRET) experiments. After 2 h of culture, the medium was replaced by 2 mL of FBS-free MEM with 2 µM blebbistatin. All experiments were performed at room temperature. Only elongated cells with clear cross striations and without granulation were used for experiments.

### 2.3. Patch-Clamp Technique

Spontaneous I_TI_, caffeine-induced NCX currents and I_Ca,L_ were measured using a perforated patch-clamp technique in freshly isolated myocytes as previously described [1]. A HEKA EPC-10 amplifier (HEKA Elektronik GmbH, Lambrecht, Germany) was used to measure I_Ca,L_ elicited by a 200 ms depolarization from −80 to 0 mV and the tail currents activated upon repolarization to −80 mV at a pacing frequency of 2Hz. The I_Ca,L_ amplitude was determined as the difference between the peak inward current and the current level at the end of a 200 ms depolarization pulse. Na^+^ currents were eliminated by a 50 ms pre-pulse to −50 mV. I_TI_ currents were measured at −80 mV and the spontaneous I_TI_ frequency was used to measure Ca^2+^ waves induced by spontaneous SR Ca^2+^ release [7]. SR Ca^2+^ load was measured using the NCX current time integral elicited by a rapid application of 10 mmol/L caffeine [16], assuming a stoichiometry of 1Ca^2+^:3Na^+^ for the NCX. Series resistance compensation was not performed. The extracellular medium (pH = 7.4) contained: NaCl 127 mM, TEA 5 mM, HEPES 10 mM, NaHCO_3_ 4 mM, NaH_2_PO_4_ 0.33 mM, glucose 10 mM, pyruvic acid 5 mM, CaCl_2_ 2 mM, and MgCl_2_ 1.8 mM. The intracellular medium (pH = 7.2 with CsOH) contained: aspartic acid 109 mM, CsCl 47 mM, Mg_2_ATP 3 mM, MgCl_2_ 1 mM, Na_2_phosphocreatine 5 mM, Li_2_GTP 0.42 mM, HEPES 10 mM and 250 μg/mL amphotericin B.

### 2.4. Confocal Calcium Imaging

Freshly isolated intact myocytes were loaded with a fluorescent Ca^2+^ dye (2.5 μM fluo-4 AM) for 20 min under control Tyrode perfusion (in mmol/L): 140 NaCl, 4 KCl, 1.1 MgCl_2_, 10 HEPES, 10 glucose, 1.8 CaCl_2_; pH = 7.4, with NaOH). Ca^2+^ imaging was carried out with and a 63x glycerol-immersion objective and a resonance scanning Leica TCS SP8 X microscope (Leica Biosystems, Wetzlar, Germany) in the frame scanning mode at a frame rate of 90 Hz, with images being acquired every 30 s. Fluo-4 (Invitrogen, MA, USA) was excited at 488 nm and fluorescence emission measured between 500 and 650 nm. Image analysis was performed using IDL software (version 7.0.6, Research System Inc., Boulder, CO, USA). Images were corrected for the background fluorescence. The fluorescence values (F) were normalized by the basal fluorescence (F_0_) in order to obtain the fluorescence ratio (F/F_0_). Analysis to identify Ca^2+^ sparks was carried out using an automated detection system and a criterion that limited the detection of false events while detecting most Ca^2+^ sparks [17].

### 2.5. Fluorescence Resonance Energy Transfer (FRET)-Based Live Cell Imaging of Intracellular cAMP

FRET measurements were performed on human atrial myocytes 48 h after transduction with the adenovirus encoding Epac1-camps [18], pm-Epac1 [19] and Epac1-JNC [20]. A K^+^-Ringer solution containing: 144 mM NaCl, 5.4 mM KCl, 1 mM MgCl_2_, 1 mM CaCl_2_, 10 mM HEPES, adjusted to 7.4 pH with NaOH at room temperature (RT) was used to maintain cells, and images were taken every 5 s. The FRET system used consisted of a standard inverted microscope (Leica DMI3000, Leica Biosystems, Wetzlar, Germany) with a 63×/1.4 oil immersion objective, an LED light source (pE-100, CoolLED), a beam splitter (DV2, Photometrics, Birmingham, UK), and a camera (CMOS camera optiMOS, Photometrics, Birmingham, UK). A single-wavelength light-emitting diode (LED, 440 nm) was used to excite the donor fluorophore (CFP), which was controlled by an Arduino digital-to-analog input-output board (Arduino). A dual view beam splitter (Cube 05-EM, 505 dcxr, D480/30m, D535/40) was used to split the emission light into donor (CFP) and acceptor (YFP) channels.

The fluorescence emitted was then measured at 475/30 nm and 527/40 nm, for CFP and YFP respectively and expressed the percent of 475/527 nm. To acquire time-lapse images the software Micro-Manager 1.4.5 [21] was used. Data analysis was performed using Microsoft Excel (version 16.4, 20121301).

### 2.6. Data Analysis and Statistics

Values were expressed as means ± SEM, and a two-tailed student’s t-test used to assess significant differences between data pairs with a significance value of *p* < 0.05. ANOVA was used for multiple effects comparison, whereas the Student–Newman–Keuls post-test was used to evaluate specific effects. Experimenters were blinded to clinical information.

## 3. Results

### 3.1. cAMP-Dependent Regulation of L-Type Ca^2+^ Current and Spontaneous SR Ca^2+^ Release Show Opposite Changes in Atrial Fibrillation

In agreement with previous studies [1,2,6], myocytes from patients with AF had a significantly smaller I_Ca,L_ density and a higher I_TI_ frequency at baseline. Furthermore, SR Ca^2+^ loading was similar for both patient groups. Next, the effects of the PKA-inhibitor H-89 (10 μM) at baseline were tested to determine if the changes in I_Ca,L_ density or I_TI_ frequency could be associated with changes in cAMP-dependent PKA signaling. In AF-myocytes I_Ca,L_ density was comparable before and after exposure to H-89, yet notably in Ctl myocytes I_Ca,L_ density was significantly reduced to levels seen in AF myocytes (Figure 1a). Inhibition with H-89 induced a small shift to the right on the current-voltage (I-V) relationship after H-89 treatment in AF myocytes (Figure 1b). The I_TI_ current frequency was practically abolished in both patient groups by H-89 (Figure 1c), although no effect was observed on Ca^2+^ sparks frequency or density (Figure 1d). Indeed, the inhibition of I_TI_ by H-89 was observed to be relatively stronger in patients with AF (7.6-fold), than Ctl ones (2.5-fold), indicating that the activation of RyR2 is more dependent on PKA at baseline in myocytes from AF patients. Jointly, these results suggest cAMP compartmentation which promoted a PKA-dependent I_Ca,L_ activation at baseline in Ctl patients, which is not seen in patients with AF, and a PKA-dependent RyR2 activation at baseline in AF patients, which could contribute, together with other factors, to the enhanced RyR2 activity in AF.

### 3.2. CaMKII Inhibition Reduces I_Ca,L_ and Ca^2+^ Sparks but Not I_TI_ Frequency

To assess whether the observed changes in intracellular Ca^2+^-handling were also modulated by CaMKII-dependent regulation, the effects of the CaMKII inhibitor KN-93 (1 μM) and its inactive analog KN-92 (1 μM) were investigated. I_Ca,L_ density was significantly reduced in response to KN-93 in myocytes from AF and Ctl patients (*p* = 0.05 and *p* < 0.005 respectively) (Figure 2a) without the I-V relationship being affected (results not shown). Neither KN-92 nor KN-93 had a significant effect on spontaneous I_TI_ frequency in the same myocytes (Figure 2b). However, CaMKII inhibition with KN-93 significantly reduced Ca^2+^ spark density and frequency in AF (*p* < 0.0001) (Figure 2c), as well as the SR Ca^2+^ load (Appendix A).

### 3.3. β-Adrenergic Stimulates I_Ca,L_ Equally in Ctl and AF Patients but Has a Relatively Smaller Effect on I_TI_ Frequency in AF

To evaluate whether β-adrenergic signaling was affected in AF, myocytes from Ctl and AF patients were exposed to the β-adrenergic agonist isoproterenol (ISO) (100 nM). As shown in Figure 3a,b, ISO promoted a relative similar increase in I_Ca,L_ density in myocytes from both groups of patients (2.7-fold in Ctl vs. 3.6-fold in AF). However, superimposed current traces before and after ISO exposure shown in Figure 3a (left panel) highlight that I_Ca,L_ amplitude in patients with AF remained smaller in absolute terms after β-adrenergic stimulation with ISO compared to Ctl (right panel). Additionally, ISO caused a significant negative shift in the I-V relationship in both groups (*p* < 0.001 in both Ctl and AF) (Figure 3b).

ISO eliminated the observed differences in I_TI_ frequency (Figure 3c) and I_TI_ amplitude (Appendix A) among Ctl and AF myocytes. This means that, in relative terms, the stimulatory effect of ISO on the spontaneous I_TI_ frequency was smaller in myocytes from patients with AF (2.1-fold) than from Ctl patients (5.5-fold).

To assess whether this effect was secondary to effects on SR Ca^2+^ loading or Na^+^-Ca^2+^ exchanger (NCX) activity, a protocol for caffeine-induced Ca^2+^ release from the SR was used to elicit the Ca^2+^ load in basal and upon ISO stimulation. Thus, the SR Ca^2+^ content was cleared with a brief caffeine pulse before reloading the SR during 30 stimulation pulses. Representative traces of caffeine-induced NCX-currents recorded before and after ISO exposure can be seen in Figure 3d. Myocytes from both patient groups showed the same ability to load Ca^2+^ into the SR before ISO treatment (Figure 3d), and similar increased reloading capability after ISO. Noticeably, the relative effect of ISO was not significantly different in myocytes from Ctl and patients with AF. The time constant (tau) for the decay of the NCX-current upon caffeine was also comparable in myocytes from Ctl and AF patients at basal and upon ISO stimulation (Figure 3e).

### 3.4. β-Adrenergic Stimulation Produces Different Increase in cAMP Levels in Ctl and AF in Different Cellular Compartments

FRET was first used to test whether the effects of ISO observed on I_Ca,L_, Ca^2+^ load, sparks and I_TI_ were linked to cAMP compartmentalized dynamics. Thus, we transduced human isolated atrial myocytes from Ctl and AF patients with genetically encoded FRET-based biosensors for cytosolic cAMP (Epac1-camps), sarcolemmal cAMP (pm-Epac1) and cAMP in the vicinity of the RyR2 (Epac1-JNC). Application of ISO demonstrated that AF promotes desensitization on the β-adrenergic pathway. Figure 4a shows a significantly smaller increase in cytosolic cAMP levels upon ISO stimulation in AF myocytes compared to Ctl myocytes (*p* < 0.0001). Interestingly, this deregulation on cAMP levels is differentially regulated in the different cellular compartments. Contrary to the reaction in the cytosol, the increase of cAMP levels upon ISO stimulation was similar between Ctl and AF myocytes in the sarcolemma and the RyR2 compartments (Figure 4a).

Next, we evaluated the role of PDEs in the control of basal cytosolic and local cAMP signaling (using the genetically encoded FRET-based sensors Epac1-camps, pm-Epac1 and Epac1-JNC) by inhibiting all PDEs expressed in the atria [22] using IBMX (100 µM) together with the PDE8 inhibitor PF-04957325 (100 nM), in atrial myocytes from Ctl and AF patients. Contrary to the effects elicited by ISO, application of IBMX+PF produced a similarly large increase in basal cytosolic cAMP levels by PDE inhibition in Ctl and in AF myocytes. However, IBMX+PF promoted differential increases of cAMP levels in Ctl versus AF depending on the compartment. Thus, in AF the increase of basal cAMP levels was twice as big as in Ctl at the sarcolemma and nearby the RyR2 (Figure 4b).

These results indicate that β-adrenergic signaling is compartmentalized by PDEs in human atrial myocytes and suggest that cAMP-dependent PKA signaling might have a prominent role regulating the phosphorylation of specific Ca^2+^-handling proteins, such as RyR2, in AF.

### 3.5. Carvedilol, but Not β_1_-Blocker Treatment, Attenuates PKA Dependent Increase of I_TI_ in AF

The above results suggest that remodeling of β-adrenergic signaling in AF patients might contribute to the observed elevation of the spontaneous I_TI_ frequency in myocytes from these patients. To further evaluate this, the effect of patients treated with β-blockers (BBs) on I_Ca,L_ density, I_TI_ frequency and SR Ca^2+^ load was explored. Comparing atrial myocytes from Ctl or AF patients who were treated with BBs and from patients who did not receive this treatment (no BBs) (Figure 5a–c), no significant difference was found on I_Ca,L_ density (Figure 5a), SR Ca^2+^ load (Figure 5b), or I_TI_ frequency (Figure 5c). Statistical analysis taking into account confounding effects of concurrent disease, risk factors and other treatments different than BBs (including ACE inhibitors, angiotensin receptor blockers and amiodarone) revealed no differences in none of the tested parameters. However, separation of BBs treatment in β_1_-blockers and Carvedilol (Figure 5d–f) revealed that myocytes from AF patients who received Carvedilol treatment had reduced I_TI_ frequency (Figure 5).

## 4. Discussion

Our study shows that AF is associated with (1) a cAMP/PKA-dependent reduction of the I_Ca,L_ density, (2) the upregulation of cAMP/PKA-dependent stimulation of SR Ca^2+^ release events large enough to produce I_TI_, and 3) PDE-dependent specific cAMP signaling in each relevant compartment for Ca^2+^-handling regulation. While inhibition of CaMKII reduced Ca^2+^ sparks frequency, it did not reduce I_TI_ frequency in myocytes from patients with AF. Furthermore, Carvedilol, but not β_1_-blockers, strongly reduced the I_TI_ frequency. Together our findings suggest that cAMP/PKA-dependent compartmentalized signaling has a potential to prevent large proarrhythmic SR Ca^2+^ release events and/or increase I_Ca,L_ density in AF and that Carvedilol, but not β_1_-blockers, may be a new therapeutic strategy to reduce Ca^2+^ release-induced atrial arrhythmias.

### 4.1. PKA-Dependent Modulation of Ca^2+^ Homeostasis in AF

In line with previous studies [2,6,23] atrial myocytes from patients with AF showed a marked reduction in I_Ca,L_ density. Although different mechanisms have been proposed to explain this reduction, i.e., reduced channel expression [24,25], redox modulation [26], or phosphatase 2-mediated dephosphorylation [11], none of them fully clarify this phenomenon. Here we show that PKA-inhibition had almost no effect on I_Ca,L_ density in AF myocytes while significantly reducing I_Ca,L_ density in Ctl myocytes (Figure 1a), suggesting that L-type Ca^2+^ channels were already dephosphorylated in AF. However, given that ISO had similar relative effects on the I_Ca,L_ density in myocytes from patients with and without AF, a down-regulation of L-type Ca^2+^-channel expression and/or changes in the redox state of the Ca^2+^ channel are also likely in AF.

PKA- [3,5,6] as well as CaMKII- [6,10] mediated RyR2 hyperphosphorylation has been associated with the increased RyR2 activity in AF patients. Voigt et al. [6] also found significantly higher cAMP levels in right atrial samples from patients with AF than those from patients in sinus rhythm. In accordance with these findings, our study found a stronger effect of the PKA-inhibitor H-89 on the I_TI_ frequency in AF than in Ctl myocytes (Figure 1c), as well as a smaller relative effect of β-adrenergic stimulation with ISO in AF (Figure 3c). Together, these results highlight the role of the cAMP-dependent PKA mediated stimulation of RyR2 activity and abnormal depolarization in AF.

Hyperphosphorylation of the RyR2 has been associated with heart failure [27,28] and treatment with BBs reduces RyR2 phosphorylation [28] and improves Ca^2+^-handling in these patients. However, in our study the I_TI_ frequency did not differ in AF myocytes from patients with or without BBs treatment (Figure 5c), suggesting that BBs treatment had no protective effect on those patients. Importantly, Carvedilol treatment had reduced I_TI_ frequency, indicating that BBs negative results were not due to a wash-out effect of the myocytes isolation. Although the selective effect of Carvedilol on the I_TI_ frequency could be due to its ability to inhibit β-adrenergic receptors [28], to modify RyR2 gating [29,30] or to prevent RyR2 oxidation [31], it is the only BB capable of reducing Ca^2+^ waves [27]. By contrast, myocytes from patients treated with B_1_Bs showed no changes in I_Ca,L_ density, SR Ca^2+^ load or I_TI_ frequency. These observations provide a mechanistic explanation for clinical studies showing that the treatment with Carvedilol, but not B_1_Bs, reduces the incidence of post-operative AF [32,33,34,35] and ischemia-induced arrhythmia [34]. In line with this, we here show that PKA inhibition effectively reduces spontaneous I_TI_, suggesting that this may be a key to preventing atrial arrhythmias where abnormal Ca^2+^ release is an underlying mechanism.

### 4.2. CaMKII-Dependent Regulation of Ca^2+^ Sparks in AF

Hyperphosphorylation of the RyR2 mediated by CaMKII has also been linked to the increased RyR2 activity seen in AF [3,6,36,37,38,39,40,41] and has been proposed as a key mechanism underlying the elevation of spontaneous Ca^2+^ release in AF. In agreement with these studies, we found that CaMKII inhibition significantly reduced the Ca^2+^ sparks frequency, although we found no significant effects on I_TI_ frequency in AF nor Ctl myocytes. Of note, none of the previous studies evaluated the effect of CaMKII inhibition in human atrial myocytes on Ca^2+^ waves frequency, only on Ca^2+^ sparks frequency, Ca^2+^ leak or RyR2 open probability. Furthermore, although CaMKII was previously highlighted as responsible for the observed effects on Ca^2+^-handling, many of those studies also found a decrease in RyR2 phosphorylation (total and at Ser2808), in incidence of spontaneous Ca^2+^ events and in the corresponding delayed afterdepolarizations when PKA was inhibited. Our findings demonstrate that a reduction of Ca^2+^ sparks or passive SR Ca^2+^ leak that had no effect on the resting membrane potential does not necessarily imply a reduction of large Ca^2+^ waves which induce a transient inward current and membrane depolarizations (Figure 2b,c). In line with this observation, it has been shown that VK-II-86, a minimally-β-blocking Carvedilol analog, reduces arrhythmias by suppressing Ca^2+^ waves and RyR2 open duration but increases Ca^2+^ sparks frequency [29]. Furthermore, knocking out phospholamban in heterozygous R4496C mice strongly increases Ca^2+^ sparks but reduces cell-wide Ca^2+^ waves preventing ventricular arrhythmia in these mice [42].

### 4.3. Key Ca^2+^-Handling Compartments Are Regulated by Specific cAMP Microdomains

The observed concurrent decrease in L-type Ca^2+^ channel phosphorylation and hyperphosphorylation of most of RyR2 in AF suggests compartmentation of cAMP/PKA signals in restricted microdomains separately regulating the phosphorylation of each Ca^2+^-handling compartment. Up to date, the majority of studies on cAMP compartmentation and the effects of PDE inhibition have been performed on ventricular cells and mostly in animal models [18,43,44,45,46,47,48], and only a few studies evaluated the contribution of cAMP/PKA compartmentation on AF pathophysiology. Increased cAMP activity [6], as well as decreased total cAMP hydrolytic activity and PDE-dependent modulation of cytosolic cAMP levels [14,15] were described in AF. Stimulation of both 5-HT [15,22,49] and adenosine A_2A_ receptors [5,50] has been linked to arrhythmias by increasing cAMP and its RyR2-dependent activation [4]. Decreased PDE activity could explain the loss in persistent AF of PDE3/PDE4 control on the propensity of 5-HT-evoked arrhythmias observed in human atrial trabeculae from patients in sinus rhythm [22]. PDE1a, PDE1c, PDE2a, PDE3a, PDE3b, PDE4a, PDE4b, PDE4d and PDE8a were found to be expressed in human atria [51]. Pharmacological inhibition of PDE4 in Ctl patients increased Ca^2+^ sparks and waves frequency, I_Ca,L_ and arrhythmias [14]. However, PDEs redistribution could increase their influence on cAMP signals in discrete Ca^2+^-handling compartments in AF. Thus, although PDE4 activity is reduced in AF, this PDE was suggested to be, at least in part, responsible for the enhanced frequency of spontaneous SR Ca^2+^ release in AF [14,52]. Moreover, PDE4 inhibition increased the incidence of arrhythmias in human atrial strips during β-adrenergic stimulation [14].

Our FRET experiments to directly measure cAMP levels in living atrial myocytes from Ctl and AF patients demonstrated that although cytosolic cAMP levels increased less in AF compared to Ctl upon ISO stimulation, they are differentially regulated in the different Ca^2+^-handling compartments (Figure 4a). Indeed, cAMP/PKA signals can reach L-type Ca^2+^ channels to the same extent in AF as in Ctl myocytes upon β-adrenergic stimulation, providing an explanation to the same relative effect on the I_Ca,L_ density upon ISO stimulation in myocytes from the two groups of patients. Furthermore, inhibition of all PDEs increased basal levels of cAMP nearby L-type Ca^2+^ channels twice as much in AF compared to Ctl (Figure 4b), suggesting that PDEs may be responsible for the dephosphorylation of the L-type Ca^2+^ channel, reducing I_Ca,L_ in AF.

At the same time, a larger increase on basal cAMP after PDE inhibition nearby RyR2 in AF (Figure 4b) would suggest an increase of PDEs in this compartment and thus, a reduction of cAMP-dependent PKA phosphorylation of these receptors. Within the hyperphosphorylated context of AF, the increase in CaMKII phosphorylation of RyR2, would maintain SR Ca^2+^ load by increasing Ca^2+^ sparks and SR Ca^2+^ leak avoiding proarrhythmic effects of Ca^2+^ waves and I_TI_, which at least at a certain point of the AF progression would need to be accompanied by an attempted restriction of the strong PKA phosphorylation by PDEs in this compartment.

## 5. Conclusions

Many conditions, such as catecholaminergic polymorphic ventricular tachycardia (CPVT), heart failure (HF), and AF, have been linked to abnormal SR Ca^2+^ release and increased phosphorylation of the RyR2 [3,5,27,28]. Additionally, BBs have been reported to reduce RyR2 phosphorylation in CPVT and HF [28,53]. Here we show that selective PKA inhibition effectively reduces spontaneous SR Ca^2+^ release in myocytes from patients with AF without affecting Ca^2+^ entry via L-type Ca^2+^ channels, therefore highlighting modulation of PKA activity as a potential key target to prevent atrial arrhythmia where abnormal Ca^2+^-release is an underlying mechanism. In line with this finding, spontaneous SR Ca^2+^ release was significantly reduced in patients treated with Carvedilol, the only BB capable of reducing SR Ca^2+^ waves [29].

In conclusion, our results show that cAMP-dependent PKA signaling is highly compartmentalized and regulated by PDEs, that PKA inhibition effectively reduces spontaneous SR Ca^2+^ release in AF, and that Carvedilol treatment reduces the incidence of spontaneous Ca^2+^ release events and I_TI_ in myocytes from patients with AF.

Furthermore, increasing evidence indicates that β-adrenergic signaling can activate different pathways other than PKA. Thus, β-adrenergic stimulation can also activate CaMKII via cAMP in a PKA-independent (activating Epac, nitric oxide synthase 1 and protein kinase G) [54,55,56] pathway, increasing Ca^2+^ sparks frequency and diminishing Ca^2+^ sparks amplitude, while decreasing SR Ca^2+^ load [54]. In line with this observation, it has been shown that Carvedilol reduces arrhythmias by suppressing Ca^2+^ waves and RyR2 open duration but increasing Ca^2+^ sparks frequency [29]. Interestingly, we found increased expression of Epac2 in AF [51], the Epac isoform which is highly concentrated at Z lines and thus implicated in SR Ca^2+^-handling [57]. In this study, we propose a model where PKA and CaMKII phosphorylation complement each other in order to adapt Ca^2+^-handling to AF remodeling progression. CaMKII would enhance RyR2 phosphorylation at the dyadic clefts during the first episodes of AF or at the beginning of the progression of the arrhythmia increasing SR Ca^2+^ leak as an attempt to reduce SR Ca^2+^ load. A similar mechanism was proposed by Zhou et al. [29], and Díaz et al. [58], as well as in response to sudden increases in heart rate, adrenergic stress or stimulation in normal hearts [59,60,61,62,63]. However, and in line with previous CaMKII studies regarding its affinity and RyR2 Ca^2+^ sensitivity [63,64,65], during diseases such as persistent AF with chronic adrenergic stimulation, enhanced cytosolic and dedicated CaMKII activity would worsen the effects of PKA-hyperphosphorylation by contributing to abnormal diastolic Ca^2+^ release from the SR, and triggering arrhythmias.

Although in atrial myocytes, L-type Ca^2+^ channels (LTCC) are equally distributed in and out of the tubular system [66], AF-associated cell hypertrophy and structural changes in the sarcolemma and RyR2 microdomains [67,68] may affect LTCC function, Ca^2+^-induced Ca^2+^-release events, RyR2 coupling to LTCC, PDE microdomains, RyR2 Ca^2+^ sensitivity and CaMKII activity. Further investigation regarding how structural changes in AF remodeling affect PKA, CaMKII and PDE interactions in basal and during β-adrenergic stimulation would provide valuable insights into AF pathophysiology.

## 6. Study Limitations

Due to the isolation process potentially washing out acute effects of BB treatment, this study could only confirm long-term effects of BBs on the remodeling of Ca^2+^ handling in atrial myocytes. However, the exclusion of patients who received concomitant treatment that theoretically could mask any BB effects did not alter our results.

Several reasons, such as differences in patient cohort or in experimental conditions, could account for the discrepancies between this and previous studies. Usage of the perforated patch-clamp technique in the present study, preserving cAMP-dependent signaling, compared to using the ruptured patch-clamp technique which has been reported to attenuate cAMP-dependent signaling and removes the effects observed on I_Ca,L_ during perforated-patch recordings [40], is likely an important factor, as any interference in cAMP-dependent signaling could bias towards CaMKII-dependent signaling. The present study used a physiological Ca^2+^ concentration (2 mM) rather than a high Ca^2+^ concentration (5 mM), commonly used to increase SR Ca^2+^ release events, and which would produce an overestimation of CaMKII-dependent signaling.

## Figures and Tables

**Figure 1 cells-10-03042-f001:**
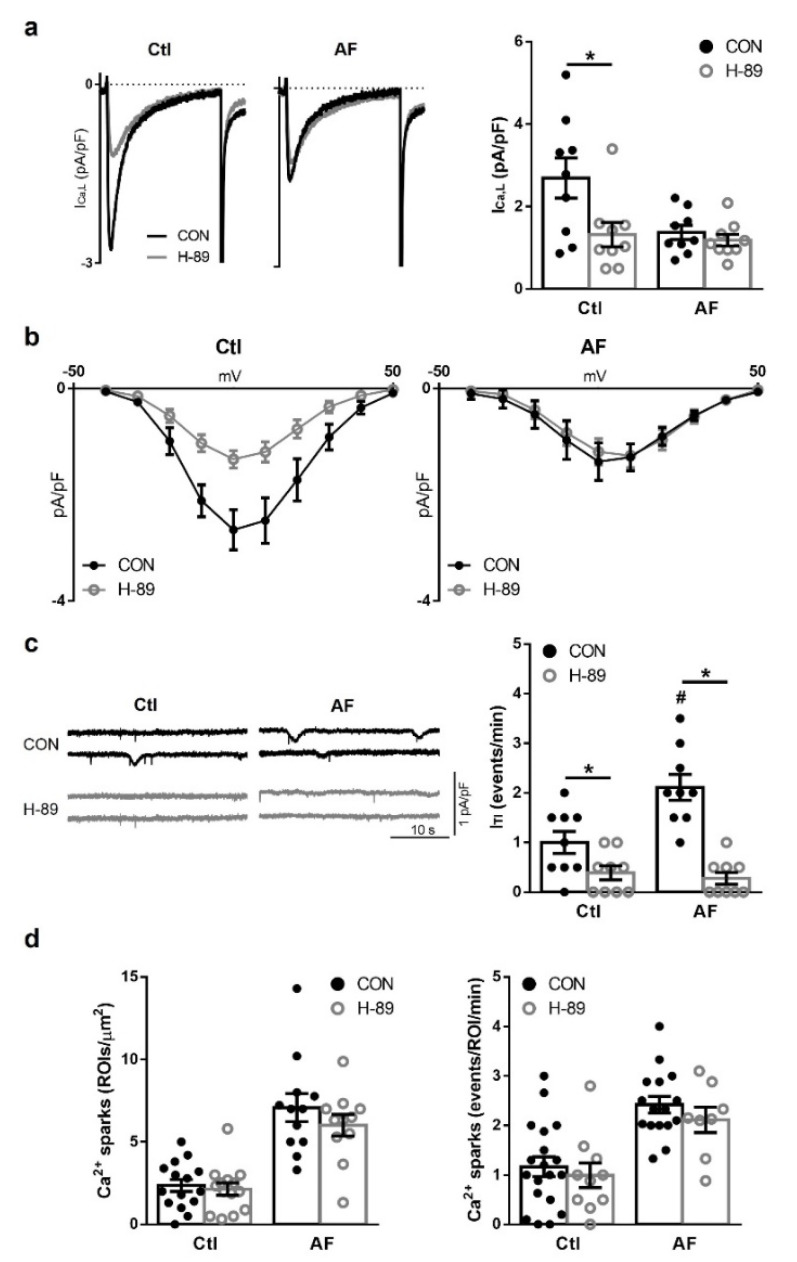
Effect of protein kinase A (PKA) inhibition on I_Ca,L_, I_TI_ and sarcoplasmic reticulum (SR) Ca^2+^ load. (**a**) (**left**): Representative I_Ca,L_ recordings in myocytes from a patient in sinus rhythm (Ctl) and a patient with atrial fibrillation (AF) before (CON, black trace) and after exposure to the selective PKA inhibitor (H-89, 10 μM, grey trace). (**right**): Average effect of H-89 in Ctl and AF patients. (**b**) Mean effect of H-89 on the current-voltage relationship in Ctl and AF patients. (**c**) (**left**): Representative traces of spontaneous I_TI_ recorded in Ctl and AF myocytes before and after exposure to H-89. (**right**): Mean effects of H-89 on the spontaneous I_TI_ frequency in Ctl and AF patients. (**d**) Average effects of H-89 on Ca^2+^ sparks density (**left**) and frequency (**right**) before and after exposure to H-89. ROI: region of interest. Significant differences between treatments are indicated with * and between groups with #.

**Figure 2 cells-10-03042-f002:**
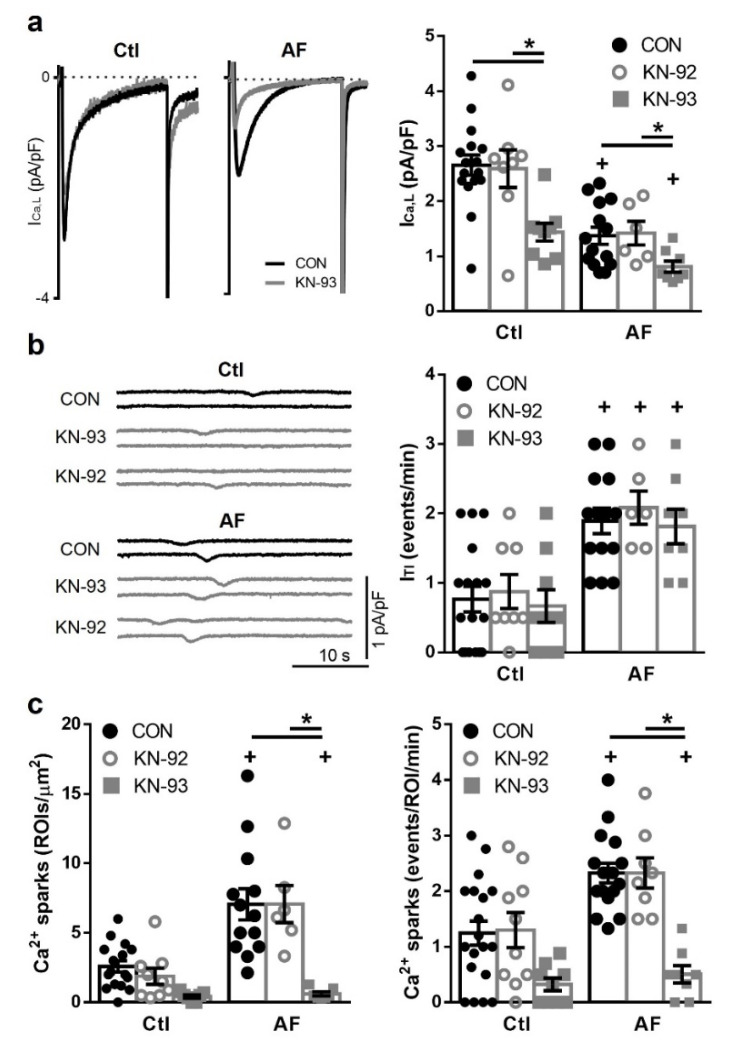
Effects of Ca^2+^/Calmodulin-dependent protein kinase II (CaMKII) inhibition on I_Ca,L_, I_TI_ and Ca^2+^ sparks. (**a**) (**left**): Representative I_Ca,L_ recordings in myocytes from a patient in sinus rhythm (Ctl) and a patient with atrial fibrillation (AF) before (CON, black trace) and after exposure to a selective CaMKII inhibitor (KN-93, 1 μM, grey trace). (**right**): Average effects of KN-92 (1 μM, the inactive CaMKII inhibitor) and KN-93 in Ctl and AF patients. (**b**) (**left**): Representative recordings of spontaneous I_TI_ before and after exposure to KN-92 and KN-93 in myocytes from in Ctl and AF patients. (**right**): Average effects of KN-92 and KN-93 on the spontaneous I_TI_ frequency in AF and Ctl myocytes. (**c**) Average effects of KN-92 and KN-93 on Ca^2+^ sparks density (**left**) and frequency (**right**) before and after exposure to KN-92 and KN-93. Significant differences between treatments are indicated with * and between groups with +.

**Figure 3 cells-10-03042-f003:**
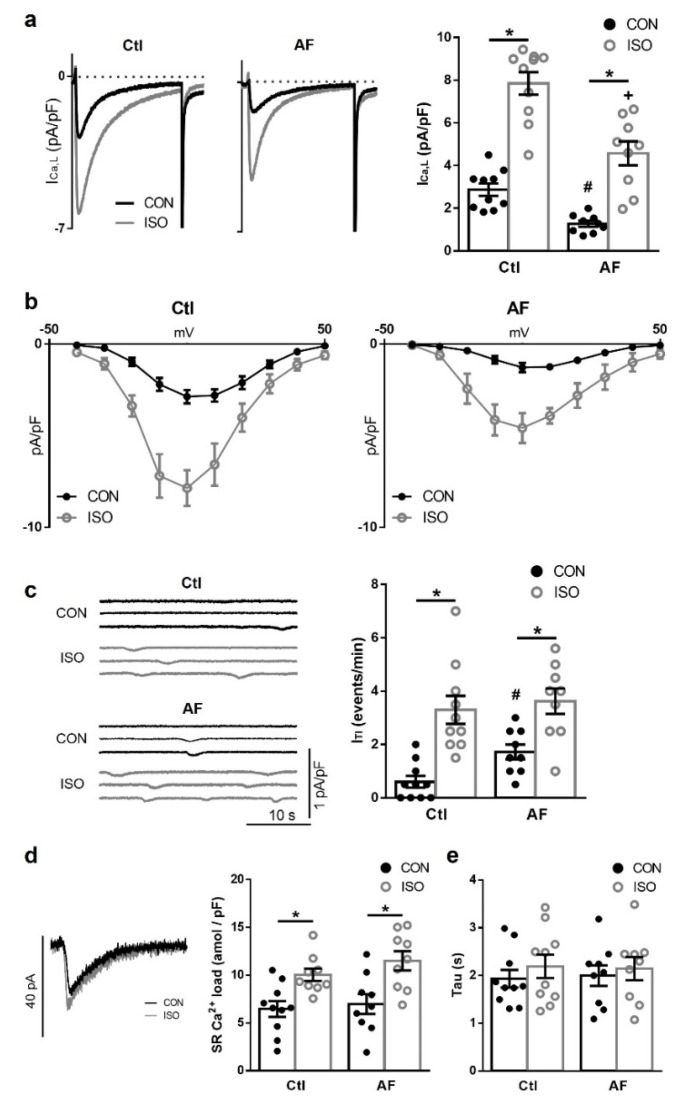
Effect of β-adrenergic stimulation on I_Ca,L_, I_TI_ and sarcoplasmic reticulum (SR) Ca^2+^ load. (**a**) (**left**): Representative I_Ca,L_ recordings in myocytes from a sinus rhythm (Ctl) patient and a patient with atrial fibrillation (AF), before (CON, black trace) and after exposure of the myocyte to 30 nM isoproterenol (ISO, grey trace). (**right**): Average effects of ISO in Ctl and AF patients. (**b**) Mean current-voltage relationship for I_Ca,L_ in Ctl and AF myocytes before and after exposure to ISO. (**c**) (**left**): Representative recordings of I_TI_ before and after ISO in Ctl and AF. (**right**): Average effects of ISO on I_TI_ in Ctl and AF myocytes. (**d**) (**right**): Representative recordings of caffeine-induced NCX-currents before and after ISO. (**left**): Average ISO effect on SR Ca^2+^ load in Ctl and AF. (**e**) Average ISO effect on time constant (Tau) in Ctl and AF. Significant differences between treatments are indicated with *, between groups with #.

**Figure 4 cells-10-03042-f004:**
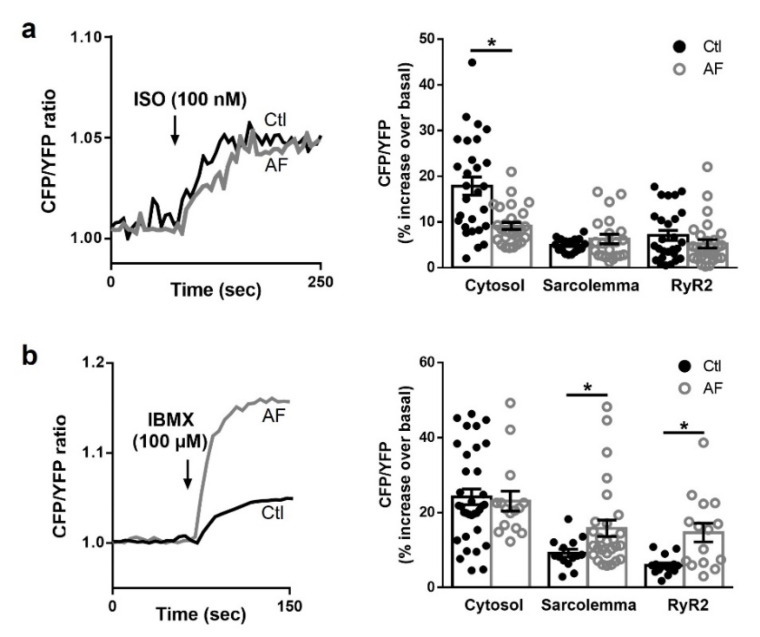
Cytosolic and local cAMP dynamics. (**left**): representative kinetics of fluorescence resonance energy transfer (FRET) changes (expressed as CFP/YFP ratio) in the ryanodine receptor (RyR2) microdomain recorded in human atrial myocytes from patients in sinus rhythm (Ctl) and with atrial fibrillation (AF), transduced with Epac1-camps-JNC sensor and treated with (**a**) isoproterenol (ISO, 100 nM) and (**b**) the non-selective phosphodiesterase inhibitor (IBMX, 100 μM) and the phosphodiesterase type 8 selective inhibitor PF-04957325 (100 nM). (**right**): average increases in cAMP levels in the cytosol and in different compartments (sarcolemma and RyR2) of living human atrial myocytes, measured as total cAMP produced after exposure (**a**) to ISO and (**b**) to the inhibitor IBMX. Significant differences between treatments are indicated with *.

**Figure 5 cells-10-03042-f005:**
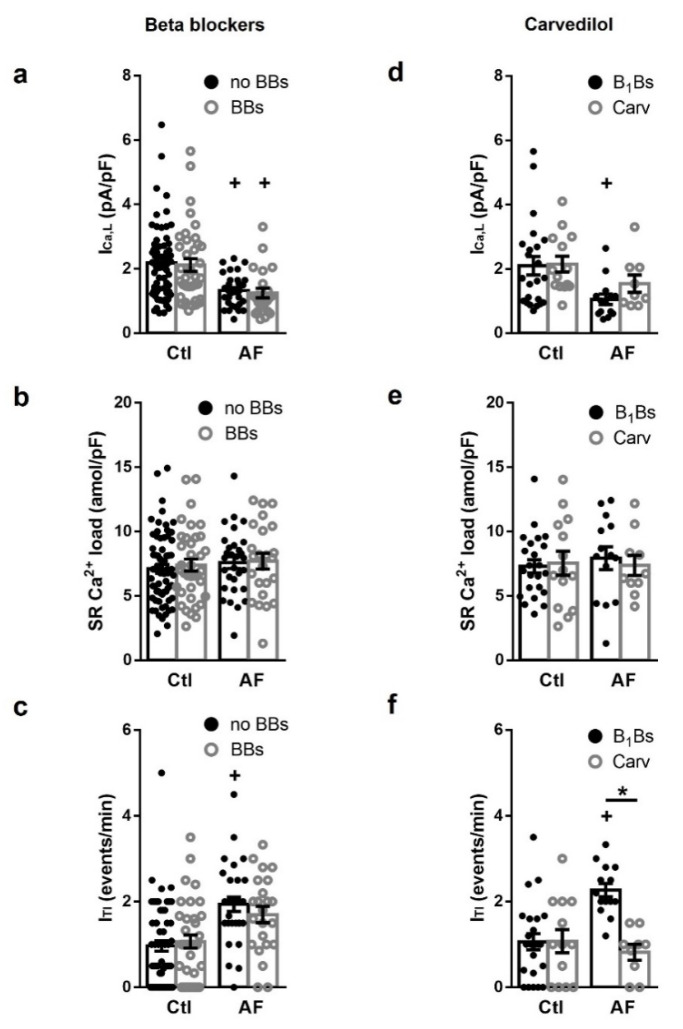
Effect of β-blocker treatment of patients on Ca^2+^-handling. (**left**): average effects of the treatment of patients with β-blockers (BBs) or without BBs (no BBs) on (**a**) I_Ca,L_, (**b**) sarcoplasmic reticulum (SR) Ca^2+^ load and (**c**) I_TI_, in patients in sinus rhythm (Ctl) and with atrial fibrillation (AF). (**right**): Carvediol effects on (**d**) I_Ca,L_, (**e**) SR Ca^2+^ load and (**f**) I_TI_, in Ctl and AF. Statistical significance was evaluated using a multivariate linear regression model adjusted for the confounding effects of common clinical factors showing a bias between BBs treatment as well as factors suspected to affect Ca^2+^-handling. Significant differences between treatments are indicated with * and between groups with +. Each point represents a patient mean value recorded in myocytes from a total of 53 patients with AF and 104 patients without AF. 23 of the AF patients and 37 of the no AF patients were treated with BBs.

**Table 1 cells-10-03042-t001:** Clinical and pharmacological details of patient included in this study. Continuous variables are presented as mean ± SD for normal-distributed data. Categorical data are given as number of patients (%).

Characteristics	Sinus Rhythm (Ctl)	Atrial Fibrillation (AF)
Patients (*n*)	148	96
Age (years)	60 ± 12	66 ± 14
Female (*n* (%))	40 (27.0)	50 (52.1)
Arterial hypertension *(n* (%))	91 (61.5)	53 (55.2)
Diabetes mellitus (*n* (%))	35 (23.7)	30 (31.2)
Dyslipemia (*n* (%))	52 (35.1)	37 (38.5)
**Indication for surgery**		
Coronary artery disease (*n* (%))	48 (32.4)	11 (11.5)
Valvular heart disease (*n* (%))	70 (47.3)	80 (83.3)
Both (*n* (%))	30 (20.3)	5 (5.2)
**Echocardiography data**		
Left atrial diameter (mm)	44 ± 9	51 ± 11
LVEF (%)	59 ± 8	58 ± 11
**Medication**		
ACE inhibitor (*n* (%))	50 (33.8)	39 (40.6)
AT_1_ receptor antagonist (*n* (%))	44 (29.7)	46 (47.9)
β-blocker (*n* (%))	31 (20.9)	48 (50.0)
Statins (*n* (%))	56 (37.8)	29 (30.2)
Amiodarone (*n* (%))	0 (0)	29 (30.2)
Diuretics (*n* (%))	32 (21.6)	66 (68.7)
Digitalis (*n* (%))	54 (36.5)	29 (30.2)
Nitrates (*n* (%))	1 (0.7)	6 (6.2)

Abbreviations: LVEF: left ventricular ejection fraction, ACE inhibitor: angiotensin-converting enzyme inhibitor, AT_1_ receptor antagonist: angiotensin II receptor type 1 blockers. N-numbers represent number of patients whose samples were used for cell isolation and afterwards on patch-clamp recording and/or confocal and/or fluorescence resonance energy transfer experiments.

## Data Availability

The data presented in this study are available on request from the corresponding author.

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
