# Peer review of "Abnormal Calcium Handling in Atrial Fibrillation Is Linked to Changes in Cyclic AMP Dependent Signaling"

_cells, 2021, doi:10.3390/cells10113042_

Round 1

Reviewer 1 Report

The present manuscript from Reinhardt et al utilises single cardiac myocytes isolated from patients with AF and those without arrhythmia to determine calcium handling responses to cAMP and PKA.  By carrying out patch clamp, FRET and calcium imaging the Authors show that there is compartmental differences with regards to control of RyRs and ICaL by cAMP. This could have repercussions for designing new treatments and does give some important insights. Although these data are of great interest to the field, i do have a number of comments/questions.

1) There is no mention of EPAC throughout the manuscript. It is well known that this PKA independent mechanism of  CAMKII phosphorylation is an additional way cAMP can control CAMKII signalling and phosphorylation. Therefore i believe there needs to be at least some commentary on this.

2) In the last paragraph of the discussion there is some ambiguity with the language used with regards to spontaneous calcium release and the transient inward current. I would first of all use the word 'large' instead of 'big' and consider rewriting these sentances as it could be miscontrued.

3) It could be useful to add percentages of patients to table 1 (in brackets after the actual numbers) for drugs/co-morbities etc.

4) I would also change the words 'cultivation' in section 2.2 to culture.

5) It would also be good to know for how long you used the cells after cell isolation in the acute experiments.

6) When investigating ITI, the Authors only seem to look at the number of occurences, and do not consider the ampitude. As, in theory, this current represents the amount of Calcium extruded, it may be more relevant to look at the integral of this current instead of purely number of occurences?

7) Panel D appears to be missing from Figure 1 showing SR Calcium. There is also no mention of these numbers in the main text, so i believe this needs to be added.

8) Why did you not look at Calcium sparks with H89 treatment as you did with the CaMKII inhibition experiments?  Also what was the SR load with CaMKII inhibition?

9) The fact that there was no changes in ITI despite alterations in the  number of calcium sparks with CaMKII inhibition, makes me think there may also be some changes either in the location of NCX or sensitivity to Calcium. Did the Authors try and do some NCX staining on these myocytes? This could add significant information.

10) Reference on line 278 needs to be put in the correct format

11) The Authors should be careful when discussing section 3.5. Just by looking at the title it may appear that these experiments were done in-vitro on the single myocytes, rather than grouping the patient populations. Maybe this needs to be rephrased

12) Also, regarding Figure 5, and all the cellular figures, was hierachical testing utilised? There are a lot of cells in figure 5, but we dont know how many are from each patient which could bias the results. Some consideration should therefore be given to this.

13) Did you also consider there could be a remodelling of beta-receptors? Previous work from the group of Gorelik and others have shown how this is the case in heart failure- so i would be interested to know if such work has been also done in atrial fibrillation.

14) I think the manuscript needs a second proof read, as there are a couple of grammatical issues in terms of tenses etc..

Reviewer 2 Report

In this study Reihardt et al. isolated myocytes from the atria of patients in sinus rhythm or atrial fibrillation. The study involved samples from a very large number of human patients which makes it scientifically and translationally important, for work of its kind. The authors discovered that LTCC function is reduced in AF myocytes and transient inward/spark activity is increased. This activity can be modulated by the inhibition of PKA and CAMKII activity. There is an apparent decrease in the effect of beta-adrenergic stimulation on calcium events in AF cells, and these are not explained by FRET experiments which look at the cAMP activation within the myocyte microdomains. However, experiments where a broadband PDE inhibitor is deployed suggest interesting remodelling of the cAMP- hydrolysis and activation in AF cells. I think that this is an important and exciting study, I have a few minor and some major suggestions.

Minor

Line 29 needs revision as the meaning isn’t clear – it seems to suggest that cAMP is depressed in AF cells, but it doesn't read like that

Table 1 needs figure legend and would be improved by the addition of percentages, there is a bit of a discrepancy between the numbers of patients per group, so this would help with understanding the make up of the cohorts.

Line 99 'appendage' not appendix

Line 145 the reference 15 should be changed to cite the following paper

https://www.ncbi.nlm.nih.gov/pmc/articles/PMC1300067/pdf/9929467.pdf which contains a much more exhaustive explanation of the analysis. This is assuming that this is still consistent with the approach? if not the process should be explained more extensively.

Line 259 vivinity   - 'vicinity'

Line 266 - the sentence beginning line 266 is poorly constructed and should be revised - it sounds like the FRET sensors are being inhibited (not the case ofc)

line 273 'differencial' – differential

Major

Line 184 - do the authors mean to say '.. indicating that the activation of RYR2 is more dependent on PKA at baseline in myocytes from AF patients??

I would contend this point as the number of Ti events is simply higher at baseline in the AF (for whatever reason). The activity drops to near zero with H89 for both sets therefore both are equally reliant on PKA. It's just that the activity starts of higher in AF.

As a result, I think that the interpretation on line 186 is a bit strong. To me the data suggest that the bar-cAMP microdomain sustains LTCC density and I-V relationship in Ctl but not AF myocytes. cAMP-PKA is also responsible for maintaining Iti event rate - but a combination of factors must be controlling the magnitude of this rate at baseline, related to AF remodelling, it may just be enhanced cAMP-PKA effects on RYR2, but this has not been demonstrated at this point in the paper.

I have a slightly different interpretation of the FRET data: hopefully, this resonates with what the authors were hoping to reflect.

Beta-AR is better able to elicit a cytosolic cAMP response in control cells (but this is rather 'epiphenomenal' as the alterations do not affect the important compartments see Bers and Zaccolo). I wonder if this is an effect of the saturating level of iso used? and not indicative of interesting physiology.

The data looking at decompartmentation (using IBMX and PF) is more interesting here, it seems that there is a greater 'PDE tone' on cAMP production in AF cells, in sarcolemmal and RyR compartments - in this scenario there is no 'cAMP' agonist pre-applied so this effect comes from receptor/AC auto-activation, this effect is higher in AF vs. Ctl - it would be fascinating to know where this effect comes from - i.e. which receptors/pathways, Is this a suppressive or compensatory effect in AF? (I note that 5-HT receptors are mentioned later in the conclusions).

Hyperphosphorylation is mentioned but no molecular studies were undertaken with this important and rare sample set can the authors comment as to why?

Why is structural compartmentation with microdomains such as t-tubules and caveolae not discussed? These compartments existing in atrial cells, albeit at a lower density that in ventricular cells. Is anything known about the structural remodelling of these cells after AF?
